# Genomic Hippo Pathway Alterations and Persistent YAP/TAZ Activation: New Hallmarks in Head and Neck Cancer

**DOI:** 10.3390/cells11081370

**Published:** 2022-04-18

**Authors:** Farhoud Faraji, Sydney I. Ramirez, Paola Y. Anguiano Quiroz, Amaya N. Mendez-Molina, J. Silvio Gutkind

**Affiliations:** 1Department of Otolaryngology-Head and Neck Surgery, University of California San Diego Health, La Jolla, CA 92093, USA; 2Gleiberman Head and Neck Cancer Center, University of California San Diego Health, La Jolla, CA 92093, USA; 3Department of Pharmacology, Moores Cancer Center, University of California San Diego, La Jolla, CA 92093, USA; sir011@health.ucsd.edu; 4Division of Infectious Disease and Global Public Health, Department of Internal Medicine, University of California San Diego, La Jolla, CA 92037, USA; 5John Muir College, University of California San Diego, La Jolla, CA 92093, USA; panguianoquiroz@ucsd.edu; 6Eleanor Roosevelt College, University of California San Diego, La Jolla, CA 92093, USA; amendezmolina@ucsd.edu

**Keywords:** head and neck squamous cell carcinoma, HNSC, HNSCC, Hippo, YAP, TAZ, FAT1

## Abstract

Head and neck squamous cell carcinoma (HNSCC) represents a highly prevalent and deadly malignancy worldwide. The prognosis for locoregionally advanced HNSCC has not appreciably improved over the past 30 years despite advances in surgical, radiation, and targeted therapies and less than 20% of HNSCC patients respond to recently approved immune checkpoint inhibitors. The Hippo signaling pathway, originally discovered as a mechanism regulating tissue growth and organ size, transduces intracellular and extracellular signals to regulate the transcriptional co-activators YAP and TAZ. Alterations in the Hippo pathway resulting in persistent YAP and TAZ activation have emerged as major oncogenic drivers. Our analysis of the human HNSCC oncogenome revealed multiple genomic alterations impairing Hippo signaling and activating YAP and TAZ, which in turn contribute to HNSCC development. This includes mutations and deletions of the FAT1 gene (29%) and amplification of the WWTR1 (encoding TAZ, 14%) and YAP1 genes (8%), together representing one of the most genetically altered signaling mechanisms in this malignancy. Here, we discuss key elements of the mammalian Hippo pathway, detail mechanisms by which perturbations in Hippo signaling promote HNSCC initiation and progression and outline emerging strategies to target Hippo signaling vulnerabilities as part of novel multimodal precision therapies for HNSCC.

## 1. Introduction

Head and neck squamous cell carcinoma (HNSCC) encompasses a heterogeneous group of malignancies arising from the upper aerodigestive tract epithelia lining the oral cavity, oropharynx, hypopharynx, and larynx. With nearly 745,000 new cases and greater than 360,000 deaths annually worldwide in 2020, HNSCC remains a major global cause of morbidity and mortality [1,2,3]. Two carcinogenic etiologies underlie most HNSCC: viral infection, primarily with human papillomavirus (HPV), and chemical carcinogen exposure via the use of tobacco products, alcohol, and betel quid [4]. Although some subtypes of nasopharyngeal carcinoma (NPC), an Epstein-Barr virus-associated cancer endemic to southern China, Southeast Asia, and North Africa, share features with virally-mediated HNSCC, it is considered a pathologically distinct malignancy, and therefore, will not be further reviewed here [5,6].

The great majority of HNSCC patients present with locoregionally advanced disease with cervical lymph node metastases and face 5-year overall survival rates ranging from 50–66% [7,8]. The prognosis for locoregionally advanced HNSCC has not appreciably improved over the past 30 years despite notable advances in surgical, radiation, and systemic oncotherapies [9]. Furthermore, while immune checkpoint inhibition (ICI) immunotherapies represent a novel treatment option for recurrent and metastatic HNSCC, the objective response rate to ICI is only approximately 20% [10,11]. These unfortunate realities underscore the need for a more detailed understanding of the biological processes underlying HNSCC carcinogenesis, progression, recurrence, and therapeutic resistance.

The Hippo pathway is a signaling cascade that integrates intracellular and extracellular signals to control cell proliferation, self-renewal, differentiation, apoptosis, and organ size [12,13]. Over the past two decades, the Hippo pathway has emerged as a tumor suppressive mechanism [14]. Comprehensive molecular characterization of cancer-associated mutations has identified alterations in the Hippo cascade and its effectors YAP and TAZ as dominant oncogenic drivers in a variety of malignancies, including HNSCC [15]. Intensive investigation of Hippo and YAP/TAZ in cancer has elucidated novel oncogenic mechanisms and has shed light on new therapeutically actionable cancer vulnerabilities [16,17,18]. In this review, we describe key elements of the Hippo pathway in mammals, detail mechanisms by which perturbations in Hippo signaling promote HNSCC initiation and progression and outline emerging strategies to target Hippo signaling vulnerabilities as part of novel precision therapies for HNSCC.

## 2. Elements of the Hippo-YAP/TAZ Pathway

The Hippo signaling pathway was initially discovered as a mechanism that restricted tissue overgrowth in *Drosophila* [13,19]. Since its discovery, Hippo signaling was revealed to be highly conserved and to play pleiotropic roles in tissue homeostasis across a diversity of organisms. These insights have led to widespread interest in understanding Hippo pathway roles and regulatory mechanisms in physiology and disease [12]. The association of Hippo pathway alterations with a variety of cancers has led to the highest interest in understanding mechanisms by which the Hippo pathway initiates tumorigenesis, aimed at the development of new targeted approaches for cancer prevention and treatment.

Multiplatform pan-cancer analyses have identified recurrent Hippo pathway alterations across the majority of cancer types evaluated [15]. Such pan-cancer analyses have shown that Hippo signaling alterations leading to YAP/TAZ activation are particularly abundant in squamous cell carcinoma (SCC). Cervical, lung, head and neck, bladder, and esophageal SCC rank among malignancies with the highest YAP/TAZ amplification frequencies, suggesting that YAP/TAZ activation represents a particularly important role in squamous cancers [15].

### The Core Hippo Kinase Cascade

The canonical model of Hippo signaling is centered on two “core” kinases and their respective regulatory proteins. The paralogs MST1 (SKT4) and MST2 (STK3) (henceforth collectively referred to as MST1/2) represent the upstream kinases of the core Hippo pathway. Diverse cellular signals activate the Hippo pathway via phosphorylation of MST1/2. In turn, MST1/2 regulates the activity of the downstream paralogous kinases LATS1 and LATS2 (LATS1/2). Phosphorylation of MST1/2 promotes its heterodimerization with SAV1, potentiating MST1/2 kinase activity to recruit and phosphorylate LATS1/2 and the LATS1/2 scaffold protein MOB1A/B [20,21]. Of note, LATS1/2 phosphorylation can also occur independently of MST1/2. MAP4K1/2/3/5, MAP4K4/6/7, and TAOK1/2/3 have been shown to directly activate LATS1/2 in an MST1/2-independent manner, thus demonstrating a partially redundant and overlapping role for MAP4Ks with MST1/2 in the core Hippo pathway [22,23]. Once LATS1/2 and MOB1A/B are phosphorylated, the activated phospho-LATS1/2:phospho-MOB1A/B complex then phosphorylates YAP (YAP1) and TAZ (WWTR1) (Figure 1) [24].

YAP and TAZ are paralogous proto-oncogenic transcriptional coactivators controlling the expression of cell proliferation, survival, self-renewal, and migration programs. Induction of YAP/TAZ-mediated transcriptional programs requires nuclear entry and interaction with DNA-binding proteins. Under physiologic conditions, the Hippo pathway regulates the cellular localization and stability of YAP and TAZ [25]. In fact, YAP/TAZ activation is regulated by a Hippo-dependent negative feedback loop, in which YAP/TAZ activation drives the expression of Hippo pathway components, such as AMOTL2, NF2, and LATS2, which in turn promote compensatory YAP/TAZ inhibition [12,26,27]. Specifically, YAP/TAZ phosphorylation promotes binding to 14-3-3 family proteins, resulting in YAP/TAZ cytoplasmic retention and functional inactivation. Furthermore, LATS1/2-induced phosphorylation can also promote further phosphorylation of cytoplasmic YAP/TAZ by casein kinase 1 (CK1), ultimately leading to YAP/TAZ ubiquitination and proteolytic degradation [24,28,29]. LATS1/2-mediated YAP/TAZ nuclear exclusion and proteolytic degradation, resulting in YAP/TAZ inhibition, represent the dominant output of Hippo pathway activation.

## 3. Upstream Hippo Regulators

Physiological organ development and growth require cells to sense and adapt to multicellular-scale environmental cues. The Hippo pathway has emerged as a signaling hub through which cells respond to the tissue environment by activating transcriptional programs (Figure 2). The Hippo pathway has been shown to cooperate with multiple tumor suppressors to sense and respond to upstream signals. These signals include cell-cell contact, cytoskeleton and cell shape, the extracellular matrix, and integral membrane receptor signaling [30,31,32]. Indeed, genetic disturbances in Hippo components in model organisms result in organ and tissue overgrowth phenotypes, suggesting the Hippo pathway is necessary to regulate cellular proliferation to maintain normal tissue architecture and organ size [33]. Acquired and inherited Hippo pathway alterations disrupt these homeostatic processes and can trigger neoplastic transformation [34,35,36].

### Direct Regulators of Core Hippo Kinases

MST1/2 phosphorylation is regulated by a variety of upstream inputs, including the PP2A complex striatin-interacting phosphatase and kinase (STRIPAK) and the tumor suppressor RASSF1A. In the setting of MST1/2 phosphorylation, STRIPAK is recruited to and binds MST1/2 via its Sarcolemma Associated Protein (SLMAP) subunit, resulting in MST1/2 dephosphorylation and inactivation [12,13]. STRIPAK dephosphorylation of MST1/2 is antagonized by the SAV1 regulatory subunit of MST1/2 as well as by RASSF1A [37,38]. STRIPAK is also involved in the regulation and inhibition of the MAP4K4/6/7 family proteins, further demonstrating a key role for STRIPAK in the regulation of the Hippo pathway [20,39]. FRMD6 is a tumor suppressor that regulates Hippo signaling by phosphorylating core Hippo kinases MST1/2 and LATS1. In addition, the *N*-terminal FERM domain of FRMD6 antagonizes YAP in the setting of FRMD6 overexpression [40]. NF2 represents an upstream activator of Hippo signaling both by interacting with AMOT to enhance LATS kinase activity and as a direct inhibitor of YAP/TAZ activity (Figure 2) [27,41,42].

## 4. Non-Hippo Signaling Inputs

### 4.1. Receptor Tyrosine Kinases, RAS, and Mitogen Activated Protein Kinases

Multiple mitogenic signal transduction pathways converge on Hippo core kinase inhibition with subsequent YAP activation. In one proposed mechanism, EGFR-RAS-MAPK signaling activation results in p-ERK-mediated phosphorylation of the AJUBA family protein WTIP. Phosphorylated WTIP demonstrates increased binding to the Hippo pathway members LATS1 and SAV1, thereby diminishing core Hippo kinase activity [43]. The receptor tyrosine kinase EGFR also has been shown to activate YAP/TAZ independently of PI3K through the direct phosphorylation of MOB1A; this is discussed in detail below (please see ‘*FAT1-Independent Mechanisms of Oncogenic Hippo Pathway Perturbation in HNSCC*’) [44].

### 4.2. G-Protein Coupled Receptors

G-protein coupled receptor (GPCR) signaling also constitutes a signaling input into the Hippo pathway and regulates Hippo activity in a class-dependent manner. G_12/13_- and G_q/11_-coupled receptors inhibit LATS1/2, while G_s_-coupled receptors activate LATS1/2 [45]. Gs-PKA signaling regulates the Hippo pathway through PKA-mediated phosphorylation of LATS1/2, which promotes YAP phosphorylation and inhibition [45]. Phosphorylation and inactivation of YAP is required for cell cycle exit and terminal differentiation, and PKA plays a role in regulating YAP activity [46,47]. Meanwhile, YAP/TAZ have been demonstrated to be required for signal transduction of the G_12/13_- and G_q/11_-coupled receptor agonist LPA and LPA-promoted cellular migration and proliferation 44. In the context of malignancy, oncogenic G_q/11_ mutant signaling requires YAP activation to drive tumor initiation and progression [48,49]. Importantly, genetic analysis of the signaling pathway by which mutant G_q/11_ activates YAP has helped identify synthetic lethal interactions between gain-of-function G_q/11_ mutations and the non-receptor tyrosine kinase FAK. This interaction has unveiled a critical vulnerability in G_q/11_ mutant cells to FAK inhibition via the disruption of YAP-activation [50].

Genetic ablation of *Gnas* or inhibition of PKA in the epidermis can result in the dramatic expansion of the stem cell compartment, whereas G_s_-PKA overactivation through the GNAS R201C mutation can drive terminal differentiation and exhaustion of the same stem cell population [51,52]. Mechanistically, this stem cell expansion is mediated in part by the loss of G_s_-PKA-mediated inhibition of YAP; without having a demonstrable effect on Wnt/β-catenin-mediated stem cell programs [51], thus demonstrating that G_s_-PKA can regulate YAP/TAZ independently of the Wnt/β-catenin signaling pathway.

### 4.3. Wnt/β-Catenin

In the absence of Wnt signaling, cytoplasmic YAP/TAZ directly binds to Axin and acts as an integral part of the β-catenin destruction complex [52]. As the main scaffold protein for the assembly of the destruction complex, Axin plays a key role in regulating Wnt/β-catenin signaling as well as YAP/TAZ subcellular localization. Axin directly binds GSK3 and YAP/TAZ, which in turn phosphorylate β-catenin and recruit the β-TrCP ubiquitin ligase, respectively. Ultimately, the formation of the phospho-β-catenin/TAZ/β-TrCP complex results in both TAZ and β-catenin proteasomal degradation. Conversely, in the presence of WNT, LRP6 competitively inhibits the binding of YAP/TAZ to Axin, releasing YAP/TAZ from the destruction complex to translocate to the nucleus, and freeing β-catenin (and TAZ) from destruction complex-mediated degradation [53].

β-catenin activity is also regulated by G_s_-PKA signaling [52]. G_s_-PKA interactions with Axin and other members of the destruction complex lead to the stabilization and activation of β-catenin [54]. This occurs through multiple mechanisms: (1) direct interaction of G_s_ with Axin to promote β-catenin activity, (2) PKA-mediated phosphorylation of GSK3 to release β-catenin from the destruction complex and allow it to translocate to the nucleus, and (3) PKA phosphorylation of β-catenin to inhibit β-catenin ubiquitination and subsequent proteasomal degradation [52,55,56]. Thus, G_s_-PKA signaling can also promote Wnt/β-catenin activity independently of its role in Hippo-YAP/TAZ regulation.

## 5. Mechanical Hippo Inputs

### 5.1. Cell Contact, Cell Shape, and the Actin Cytoskeleton

Changes in cell shape and cytoskeletal organization induced by cell density signaling can be transduced through the Hippo pathway thereby controlling YAP/TAZ subcellular localization. Contact inhibition describes an arrest in cell proliferation that occurs when cells reach high cell density. High cell density induces LATS1/2 kinase activity with subsequent YAP/TAZ phosphorylation, resulting in YAP/TAZ interaction with 14-3-3 and YAP/TAZ cytoplasmic sequestration [24,57]. The YAP/TAZ:14-3-3 complex limits YAP dephosphorylation by PP2A phosphatases. Removal of cell-cell contacts relieves PP2A inhibition, allowing for YAP-dephosphorylation, nuclear translocation, and activation of transcriptional programs that drive cell cycle re-entry. YAP activation in cancer is observed in cells that have lost cell contact sensing mechanisms, indicating that relief from contact inhibitory sensing abolishes Hippo signaling and unleashes YAP oncogenic function [24].

In parallel, at high cellular density, individual cells display reductions in the substrate-adherent cell surface area and decreased cytoskeletal tension. Such conditions result in the inhibition of Rho-GTPase activity, reduction of F-actin stress fiber formation, and YAP/TAZ inactivation. Rho- and F-actin polymerization-mediated regulation of YAP/TAZ subcellular localization may occur through Hippo-dependent mechanisms, including Rho:LIMD1-mediated inactivation of LATS1/2 (see below) [58]. In the absence of YAP-mediated cell cycle progression, the quiescent cell state can be reversed through induction of mechanical stretch, which induces E-cadherin to release YAP from the plasma membrane, enabling YAP nuclear translocation to activate cell cycle re-entry [59].

YAP/TAZ activity can be regulated by multiple Hippo pathway-independent mechanisms. Several proteins, including α-catenin, AMOT, PTPN14 and CDK1 can inhibit YAP/TAZ by directly binding and sequestering them in the cytoplasm [24,33,60]. Adherens junctions and tight junctions constitute two distinct cell-cell adhesion complexes implicated in Hippo activation. At adherens junctions, E-cadherin dimerization recruits α- and β-catenin to induce MST1/2 phosphorylation in conjunction with tumor suppressors Kibra (WWC1) and NF2 [61]. In epidermal epithelia, such as those of the upper aerodigestive tract, adherens junctions function in sensing epidermal cell density and restricting basal cell proliferation. In this regard, α-catenin is involved in sensing cell density and exerting a negative regulatory effect on YAP in cell-dense conditions. Given its role in YAP inactivation in the setting of contact inhibition, preventing inappropriate cellular proliferation, α-catenin can be considered a tumor suppressor [57]. At tight junctions, the angiomotin (AMOT) complex meditates contact inhibition by (1) directly binding and sequestering YAP at the plasma membrane and (2) activating LATS1/2 kinase activity through the tumor suppressor Merlin (NF2) [62].

### 5.2. The Extracellular Matrix

The extracellular matrix (ECM) acts both as a scaffold for cell adhesion and spread as well as a source of physical cues that influence cellular growth and survival. ECM stiffness is one of the principal determinants of cellular attachment and spreading. As such, it also plays roles in cellular growth, migration, and differentiation; in part through alterations in Hippo pathway signaling and YAP/TAZ subcellular localization.

Cellular attachment to the ECM increases nuclear YAP through activation of signaling pathways including FAK-Src, Integrin-PI3K, and Rho-GTPases [63,64,65]. Increased mechanical strain leads to JNK-mediated phosphorylation of the AJUBA family protein LIMD1. Phosphorylated LIMD1 can bind to and inactivate LATS1/2, decreasing LATS1/2-mediated phosphorylation of YAP and promoting YAP nuclear translocation [58].

Regulation of YAP/TAZ subcellular localization by ECM rigidity is also mediated by changes in cell shape and cytoskeletal tension. As noted above, F-actin polymerization and Rho-GTPases play an integral role in the regulation of YAP/TAZ translocation by mechanotransduction. RhoA GTPase activity promotes actin polymerization and stress fiber formation, thereby altering cell shape and adhesive surface area. Disruption of F-actin decreases nuclear YAP. Hippo-mediated transduction of mechanical forces into transcriptional programs of cell growth and differentiation is proposed to function as a checkpoint during stem cell development and may have implications for cancer stem cells [66].

Tumor tissue is often characterized by stiffness attributable to higher collagen expression and increased collagen cross-linking [67]. Increased lysis oxidase (LOX) enzyme activity in tumor tissue creates increased numbers of cross-links between collagen and other ECM components [68,69]. Tumor cell proliferation, differentiation, migration, and invasion are influenced by the rigidity of the tumor tissue. Given the role of Hippo signaling in sensing ECM stiffness, such conditions are likely to induce YAP activation, which in turn mediates tumor cell proliferation, differentiation, migration, and invasion programs attributed to ECM-stiff tumors [68,70].

## 6. Functions of Hippo Pathway Effectors: YAP and TAZ

### 6.1. Proliferation

Of direct relevance to squamous carcinogenesis, YAP has been demonstrated to promote the proliferation of progenitor cells in the epidermis [57,71]. Transgenic mouse models carrying an active YAP allele (YAP^S127A^) that is not subject to regulation by cytoplasm-restricting phosphorylation have demonstrated that uncontrolled YAP activation drives the expansion of proliferating basal epidermal cells of the skin and oral cavity. Conversely, knockout of YAP resulted in failure of normal skin development and loss of epidermal barrier function. TEAD family DNA binding proteins represent the major transcriptional partners for YAP in the context of epidermal development, and TEAD cellular distribution mirrors that of YAP activation, with cells exhibiting the highest proliferative activity and nuclear YAP localization also exhibiting strong TEAD expression [57].

Chromatin immunoprecipitation analyses have informed the landscape of genome-wide YAP/TAZ and TEAD binding, revealing that TEAD4 binding sites overlap with 78% of YAP/TAZ sites. Surprisingly, the overwhelming majority (89%) of YAP/TAZ/TEAD binding was found to occur at active enhancer elements marked by histone H3 monomethylation at lysine 4 residue (H3K4me1) and acetylation of lysine 27 (H3K27ac), while very few YAP/TAZ/TEAD peaks (3.6%) occurred at promoters. Greater than a third of the YAP/TAZ/TEAD direct target genes constituted a cell-proliferation program of essential factors involved in replication licensing, DNA synthesis and repair, transcriptional regulators of the cell cycle, cyclins and their activators, and factors required for completion of mitosis [72]. Taken together, these findings suggest that the YAP/TAZ/TEAD complex functions at distal enhancers to activate a transcriptional program to induce cell cycle progression. Indeed, depletion of YAP/TAZ or TEAD resulted in a severe reduction of cells in the S and G2/M phases of the cell cycle and an expansion of cells arrested in G1, demonstrating the necessity of YAP/TAZ in proliferation through interactions with TEAD family proteins [72].

### 6.2. Stemness

YAP/TAZ have been implicated as regulators of stem cell maintenance in multiple systems. YAP is elevated during induced pluripotent stem cell reprogramming and embryonic stem cells (ES) lose their pluripotent potential with YAP depletion. Embryonic stem cell differentiation is prevented by ectopic YAP expression. Genome-wide analysis helped to elucidate the role of YAP in ES differentiation, demonstrating that many YAP-inducible genes contain TEAD-binding sites that are bound by YAP and that YAP regulation of ES stemness occurs through YAP-mediated transcriptional activation of programs of ES pluripotency and self-renewal, including Sox2, Nanog, and Oct4 [73].

In the epidermis, enforced YAP activation disrupts normal epidermal stratification by both promoting expansion of the progenitor cell compartment into the suprabasal epidermal layers and preventing differentiation of these cells [57]. By depleting YAP/TAZ in cells grown at different densities and on substrates that provided different mechanosensory inputs, it was demonstrated that YAP/TAZ depleted cells exit the cell cycle and concomitantly lose expression of basal/stem cell markers while gaining expression of terminal differentiation markers; YAP/TAZ is excluded from the nuclei of differentiated cells. Conversely, ectopic expression of YAP^5SA^, which lacks inhibitory phosphorylation sites, prevents epidermal cell differentiation regardless of cell density or substrate composition [25].

In contrast to YAP/TAZ activity, Notch signaling is enhanced by low mechanical strain as experienced by cells grown on soft substrates, at higher cell density, or in the context of F-actin inhibition [25]. Inhibition of Notch signaling through various methods consistently prevented YAP/TAZ depleted cells from undergoing differentiation [25]. The opposing actions of YAP/TAZ and Notch in epidermal progenitor cell differentiation were recapitulated in vivo. YAP/TAZ are transcriptional co-activators that typically control transcription through binding enhancers distant to the transcription start site. Mechanistically, YAP/TAZ antagonizes Notch signaling ‘in cis’ through transcriptional activation of Notch inhibitors, including Delta-like ligands (DLL), promoting maintenance of epidermal stemness and inhibiting differentiation [25].

## 7. The Landscape of Hippo Pathway Alteration in HNSCC

Genome-wide molecular profiling efforts have provided detailed mutational landscapes of cancer and identified *TP53* (72%), *CDKN2A* (54%), *PIK3CA* (35%), *FAT1* (29%), and *NOTCH1* (21%) as the most frequently altered genes in HNSCC [74]. Moreover, these and our efforts have revealed Hippo pathway alterations as recurrent events in HNSCC [15,75]. The top ten most frequently altered Hippo pathway genes are mutated or altered in 61% (309/504) of patients with HNSCC and, among Hippo pathway altered HNSCCs, nearly half (47%, 144/309) possess two or more alterations in these commonly mutated Hippo genes. Consistent with their roles in oncogenesis, tumor suppressive Hippo genes predominantly undergo inactivating mutation and deletion, while oncogenic Hippo genes show amplification (Figure 3) [75,76,77]. These observations support the emerging importance of Hippo function in the upper aerodigestive tract epithelia and indicate that disturbances in the Hippo pathway may contribute to HNSCC carcinogenesis and progression.

## 8. YAP/TAZ Activation in HNSCC

HNSCC ranks among the malignancies in which YAP and TAZ amplification is most frequently observed [15]. Together YAP and TAZ are amplified in 19% of HNSCC, suggesting that a subset of these tumors are dependent on YAP/TAZ hyperactivation. Accordingly, a genome-wide CRISPR-Cas9-based inactivation screen identified dependencies on YAP or TAZ in 13 of 21 (62%) of HNSCC cell lines [78]. Interestingly, YAP/TAZ non-dependent cell lines were found to be sensitive to combined YAP and TAZ knockout, suggesting that these cell lines could compensate for either YAP or TAZ loss via its cognate paralog [78]. Consistent with this notion, others have shown compensatory upregulation of TAZ upon YAP knockdown and vice versa [79]. Taken together, these findings demonstrate the translationally important insights that a large fraction of HNSCC display a dependency on YAP and/or TAZ, and that the vast majority of HNSCC is susceptible to combined YAP and TAZ inactivation.

## 9. Mechanisms of YAP/TAZ Activation in HNSCC

### 9.1. FAT1: A Membrane-Associated Proto-Cadherin Assembling the Hippo Signalome

Although dysregulation of the Hippo pathway occurs frequently in multiple human malignancies, alterations in the canonical core Hippo kinases are rarely observed. In contrast, with an alteration rate of nearly 30%, *FAT1* mutation constitutes one of the most frequent mutations in HNSCC [75]. Inactivating *FAT1* mutations, primarily in the form of homozygous deletions, but also in the form of protein-truncating nonsense mutations, strongly suggest that FAT1 functions as a tumor suppressor [75,80,81].

Under physiological conditions, FAT1 serves as a membrane-associated scaffold to organize the activated core Hippo signaling complex (signalome). Through loss of function, ectopic expression, and protein interaction experiments, TAOK1/2/3 were demonstrated to represent the upstream kinases triggered by FAT1 to activate the core Hippo complex [75]. In this context, the FAT1 intracellular domain served to assemble the multimeric Hippo core signalome consisting of phospho-MST1 (p-MST1), p-MST2, p-SAV1, p-LATS1, and p-MOB1, culminating in YAP phosphorylation. The degree to which Hippo-mediated YAP phosphorylation is dependent upon FAT1 is underscored by the functional loss of *FAT1*, which results in the disassembly of the Hippo signalome with consequent YAP activation, nuclear localization, and activation of YAP-mediated proliferative, survival, and tumorigenic transcriptional programs. These mechanistic observations were validated using cell-based functional assays that demonstrated FAT1 expression abrogated YAP-activated transcription and cell proliferation, which was reversed upon co-expression of a Hippo-insensitive YAP mutant [75]. This study was the first to mechanistically link *FAT1* disruption with Hippo signaling disruption and uncontrolled YAP activation. Interestingly, *FAT1* mutation has been identified in carcinogen-induced models of HNSCC, potentially linking carcinogen-induced mutation and Hippo pathway perturbation as mediators of HPV-negative HNSCC initiation [82]. *FAT2*, *FAT3*, and *FAT4* are also recurrently altered in human HNSCC and in carcinogen-induced murine models of HNSCC [82,83]. However, the biological significance of these alterations remains to be examined [84]. While inactivating *FAT1* mutations are uncommon in HPV-positive HNSCC [85], preliminary analyses suggest that Hippo pathway perturbation and YAP/TAZ activation are also widespread in HPV-positive HNSCC (not shown), thus representing an area of active investigation.

### 9.2. FAT1-Independent Mechanisms of Oncogenic Hippo Pathway Perturbation in HNSCC

Independent of *FAT1* mutation, YAP activation is a prevalent feature of HNSCC. Epidermal growth factor receptor (EGFR) activation also represents a common event in HNSCC that is correlated with aggressive disease [86,87]. Recent studies have described a series of mechanisms through which EGFR and receptor tyrosine kinase downstream signaling cascades via RAS-MAPK and phosphoinositide-3-kinase (PI3K) can drive oncogenic YAP activation.

Amplification and gain of function mutations in *PIK3CA* result in the hyperactivation of its gene product PI3K and PI3K signaling pathway activation in HNSCC. PI3K signaling has been shown to induce nuclear YAP localization and YAP-target transcription in response to EGFR or GPCR agonists. Demonstrating strict signaling through PI3K and PDK1, this response was attenuated by PI3K and PDK1 inhibitors and siRNA mediated knockdown. Mechanistically, in cell contact-inhibited conditions, PDK1 binds to MST1 and LATS1 through SAV1, enabling Hippo activation and resulting in YAP phosphorylation and cytoplasmic retention, and cellular growth arrest. In the presence of growth factors or constitutive PI3K signaling, PDK1 is recruited to the plasma membrane, resulting in the dissociation of the MST1:SAV1:LATS1 complex, YAP nuclear accumulation, and cell cycle entry [88].

More recently, EGFR has been shown to activate YAP/TAZ independently of PI3K through the direct phosphorylation of MOB1A in HNSCC cells [44]. EGFR phosphorylates novel MOB1A tyrosine residues in the presence of epidermal growth factor, resulting in reductions in p-LATS1/2 and p-YAP, triggering nuclear translocation of YAP and activation of YAP-mediated transcription programs. Furthermore, inhibition of EGFR in HNSCC cells with erlotinib was shown to be sufficient in abrogating YAP activation and the expression of YAP-mediated transcriptional programs. While further studies are required to delineate the mechanism by which EGFR-mediated MOB1A phosphorylation prevents LATS1 activation, this study unveiled a novel, therapeutically actionable regulatory signaling node by which multiple receptors and non-receptor tyrosine kinases may converge with the Hippo pathway to control YAP/TAZ activity [44].

Disruption in core Hippo signaling has been shown to induce HNSCC in model systems. Genetic deletion of *Mob1a* and *Mob1b* in murine lingual epithelia led to rapid squamous cell carcinoma. Consistent with the canonical Hippo core signaling model, knockout of *Mob1a/b* in tongue epithelia resulted in diminished LATS1 but not MST1 protein, increased nuclear YAP, and upregulation in YAP target genes [89]. Much remains to be understood about core Hippo signaling and the functions of MOB1 in YAP/TAZ regulation. As illustrated in the setting of *Mob1a/Mob1b* knockout in tongue epithelia, knockout of *Yap* prevented carcinogenesis induced by *Mob1a*/b deletion, but knockout of *Taz* in the same context resulted in more aggressive SCC with more invading lesions [89]. While conditional deletion of *Mob1a/b* suggests that disruption of these core Hippo complex proteins quickly leads to carcinogenesis, *Mob1a*/*b* can also contribute to signaling cascades beyond Hippo. For example, *Mob1a* and *Mob1b* stimulate the activity of non-Hippo kinases including STK38 (NDR1) and STK28L (NDR2) [90]. While NDR1/2 have been implicated in Hippo signaling and reported to phosphorylate YAP [91], NDR1 also serves to attenuate mitogen activated kinase signaling as a negative regulator of MAP3K1/2 [92]. Indeed, collectively NDR1/2 exert pleiotropic potentially proto-oncogenic influences on cell cycle progression, apoptosis, stress signaling, and autophagy [93]. Exploring the complex interactions across Hippo and other signaling cascades may clarify seemingly paradoxical effects observed upon the genetic depletion of kinase-regulatory scaffolds, such as *Mob1a/b.*

## 10. Consequences YAP/TAZ Activation in HNSCC

### 10.1. Cancer Stemness

Chromosome 3q25-26 is a frequently amplified locus in HNSCC that harbors multiple cancer-associated genes including *WWTR1 (TAZ)*, *PIK3CA*, and *SOX2*. Co-occurrence of 3q25-26 amplification and *TP53* mutation is associated with poor prognosis in HNSCC [94]. Interestingly, a TAZ:TEAD4 complex has been shown to co-activate *SOX2* transcription in HNSCC. In this context, *SOX2* expression was found sufficient to promote cancer stem cell marker expression, tumor cell self-renewal in vitro, and tumor growth in vivo [95]. Importantly, knockdown of TAZ diminished these phenotypes and exogenous expression of SOX2 upon TAZ knockdown was sufficient to rescue them. Together, these findings show that TAZ- and TEAD4-mediated coactivation of *SOX2* expression is sufficient to induce stemness programs in HNSCC and offer the possibility that TAZ activation may mediate transcriptional programs that lead to the poor prognosis associated with 3q25-26 copy gain. In line with these findings, an independent study found that TAZ activation promotes migration, invasion, and survival by an epithelial to mesenchymal transition-like program in HNSCC cells, which in turn promotes cancer stem cell maintenance and expansion [96].

The recurrently amplified segment of chromosome 3q also harbors two genes that are frequently co-amplified in HNSCC: *TP63*, a key regulator of epidermal cell differentiation and proliferation [97], and *ACTL6A*, a subunit of the SWI/SNF ATP-dependent chromatin remodeling complexes [98]. Interestingly, p63 forms a complex with ACTL6A and other SWI/SNF subunits in HNSCC to control a stem-like transcriptional program that enhances the regenerative potential of HNSCC cells in vitro, and promotes a pro-tumorigenic proliferative, undifferentiated cancer cell state in vivo. Importantly, p63 and ACTL6A were found to directly repress transcription of *WWC1* [99]. The gene product of *WWC1*, Kibra, is a tumor suppressor that promotes Hippo pathway activation [100,101], thereby releasing Hippo-mediated inhibition of YAP/TAZ. Indeed, depletion of p63 or ACTL6A, increased Kibra expression, increased p-YAP, depressed nuclear translocation of YAP, and diminished HNSCC regenerative potential in vitro [99].

### 10.2. Tumor Progression and Poor Prognosis

Progressively increased YAP activation is a feature of oncogenic progression in diverse malignancies [102]. In the squamous epithelia of the upper aerodigestive tract, physiologically nuclear YAP is restricted to the basal epidermis [75,103]. The extent and degree of nuclear YAP have been shown to progressively increase with worsening histological severity in oral premalignant lesions and culminate with diffuse, strong nuclear YAP expression in the majority of cells in HNSCC. Furthermore, the frequency of nuclear YAP-positive tumor cells also increases with worsening histological grade in HNSCC [75,79]. Array-based transcriptome profiling upon YAP/TAZ knockdown in HNSCC cell lines has elucidated YAP/TAZ-mediated transcription programs in HNSCC. The resultant YAP/TAZ-regulated gene signature, defined as the set of transcripts downregulated upon YAP/TAZ knockdown, showed enrichment for stemness, self-renewal, cell cycle progression, and invasion programs [79]. Consistent with these findings, it has also been shown that nuclear YAP and TAZ in HNSCC are enriched at the tumor invasive front [104,105]. In addition, increased nuclear YAP expression in primary tumor specimens has been associated with lymph node metastasis, further suggesting that YAP may play a role in driving invasion and metastatic programs [105]. In line with these findings, TAZ activation was also shown to be a prevalent feature in HNSCC cell lines and an independent prognostic factor for disease-free and overall survival in patients with tongue HNSCC [106].

Whether cancer associated mutations modify prognosis or therapeutic sensitivity are key questions that may shed insight into mechanisms of tumor initiation and progression, and aid in the identification of prognostic, diagnostic, and therapeutic biomarkers. In this regard, the potential roles of YAP/TAZ on prognosis and other clinicopathologic features in HNSCC have been evaluated. YAP copy number alteration (CNA) is observed in 8% of HNSCC. While YAP amplification is associated with increased YAP transcript abundance, augmented YAP expression is often observed in HNSCC in the absence of CNA [107]. Post-translational signaling cascades regulate YAP activation; therefore, YAP activity is likely not directly related to its transcriptional expression levels. Eun and colleagues interrogated the Cancer Genome Atlas (TCGA) database to identify transcripts correlated with YAP transcript expression and CNA in order to develop a gene expression signature for YAP activation [107]. This YAP activation signature was found to stratify prognosis with regard to disease-free survival, disease-specific survival, and overall survival in four independent patient cohorts. Importantly, the YAP activation signature stratified survival after multivariable correction for confounding clinical factors known to be associated with survival (age, tumor classification, nodal classification, TNM staging group, and anatomic site) [107].

Analyses of clinical data-linked, genome-wide databases have identified interactions between PIK3CA and YAP activation in HNSCC and demonstrated that these interactions may be associated with survival outcomes. While neither mutation nor CNA in PIK3CA was associated with recurrence-free survival (RFS), PIK3CA mRNA expression was found to be associated with RFS in TCGA and an independent dataset on single and multivariable analyses [108]. Consistent with mechanistic models in which PIK3CA activation induces downstream YAP dephosphorylation and nuclear translocation to drive YAP-mediated transcription programs, HNSCC tumor samples with high PIK3CA expression displayed lower abundances of p-YAP and PIK3CAhigh tumors expressed transcriptional programs enriched for YAP/TAZ target genes [108]. These provide translational relevance to the mechanistic studies linking PIK3CA activity to YAP/TAZ activation, suggest that the PIK3CA-YAP axis may drive aggressive forms of HNSCC, and provide a strong rationale to target YAP in PIK3CA^high^ HNSCC.

### 10.3. Therapeutic Resistance

Combined with radiotherapy and surgical resection, cisplatin-based chemotherapy is a mainstay of curative-intent therapy for HNSCC [109]. Limited evidence suggests that short interfering RNA-mediated YAP-knockdown re-sensitizes cisplatin-resistant HNSCC cell lines to cisplatin. These findings suggest that YAP may be considered a potential therapeutic target for cisplatin-resistant HNSCC [110]. Using a similar line of experimentation, an independent group identified a ribosome-binding protein that mediates the interaction between the ribosome and the endoplasmic reticulum membrane, RRBP1, as a mediator of cisplatin resistance in HNSCC by augmenting YAP expression. In this setting, the use of a putative RRBP1 inhibitor, Radelozid, diminished YAP expression and sensitized HNSCC cells to cisplatin in both in vitro and in vivo assays [111]. In addition, TAZ overexpression has been shown to enhance resistance while YAP or TAZ knockdown sensitizes HNSCC cell lines to cisplatin and fluorouracil [96,112].

More recently, precision therapeutic targeting of the MAPK pathway, which is commonly activated in HNSCC and is associated with more aggressive tumor growth, nodal metastasis, and recurrence, has been evaluated in clinical trials [113,114,115]. In a window-of-opportunity trial in previously untreated HNSCC patients, the small molecule MEK inhibitor trametinib was shown to inhibit the MAPK pathway in 33% of patients, as evaluated by tumor p-ERK abundance, and to result in clinical tumor response rates in 65% of patients [116]. Given that single-modality targeting of MAPK signaling has not shown durable efficacy in a wide variety of tumor types [117], follow up studies have explored mechanisms of trametinib resistance in HNSCC that could lead to multimodal precision therapeutics with durable efficacy. In this regard, trametinib-resistant HNSCC cell lines were generated by culturing cells in sequentially increasing doses of trametinib. The resultant cells demonstrated at least 10^5^-fold increased resistance to trametinib and elevated YAP activity as measured by transcriptional reporter assays and upregulation of canonical YAP-target genes. In addition, trametinib-treated patient-derived xenografts that escaped growth inhibition were found to have 15-fold greater abundances of unphosphorylated YAP protein and upregulated YAP-target genes. Accordingly, combining trametinib with verteporfin, which inhibits YAP signaling, showed synergistic effects. These results highlight the potential importance of YAP activation in mediating resistance to MAPK pathway inhibition and suggest YAP inhibition as a potential strategy to enhance the efficacy of MAPK pathway inhibition in patients with HNSCC [118].

## 11. YAP/TAZ Activation Exposes HNSCC Vulnerabilities and Therapeutic Opportunities

Accumulating evidence demonstrating YAP and TAZ as oncogenic effectors and essential cancer dependencies in HNSCC supports the development of novel oncotherapeutics against YAP/TAZ [17,18]. However, therapeutic targeting of transcription factors remains a major challenge. Beyond YAP and TAZ, numerous cancer-associated transcription factors have emerged as promising therapeutic targets [119]. Yet the development of specific small molecule therapeutics against transcription factors has been hampered by hurdles related to the transcription factors not typically possessing enzymatic activities and practical challenges of disrupting protein:protein and protSein:nucleic acid interactions [119].

The requirement for YAP/TAZ to interact with TEAD family DNA binding proteins present an opportunity for its inhibition. An in vitro functional screen in mammalian cells for small molecule inhibitors of YAP:TEAD based transcriptional transactivation identified three porphyrin derivatives protoporphyrin IX, hematoporphyrin, and verteporfin as inhibitors of YAP-induced transcription [120]. Coincidentally, an independent screen identified these same three porphyrins as disruptors of YAP:TEAD interaction in *Drosophila*. Further studies validated verteporfin as a strong inhibitor of YAP:TEAD interaction [120]. Verteporfin is an FDA-approved photosensitizer used in photodynamic therapy for the treatment of age-related macular degeneration, suggesting verteporfin exerts cellular effects beyond the inhibition of YAP:TEAD interactions (Figure 4) [121]. Despite being studied for the past decade as a YAP inhibitor, structural mechanistic details about its interaction with YAP/TAZ remain unclear, and its effects on cellular targets other than YAP/TAZ, including autophagosome inhibition, suggesting that verteporfin constitutes a nonspecific YAP/TAZ inhibitor [122].

Further study of the YAP/TAZ:TEAD interaction has revealed a unique mechanism through which YAP/TAZ and TEAD family proteins interact to co-activate transcription and has unveiled a novel opportunity for pharmacologic inhibition. YAP/TAZ and TEAD complex formation require an autopalmitoylation event in which TEAD catalyzes the covalent attachment of a palmitate fatty acid to itself. TEAD autopalmitoylation is both necessary for YAP/TAZ:TEAD complex formation and YAP/TAZ-mediated transcriptional co-activation [129]. Small molecule inhibitors of TEAD autopalmitoylation have recently been demonstrated to bind TEAD proteins and prevent TEAD autopalmitoylation, disrupt YAP/TAZ interaction with TEAD proteins, diminish YAP/TAZ:TEAD transcriptional co-activation, downregulate YAP/TAZ:TEAD target transcripts, and inhibit in vivo tumor growth of *NF2*-deficient cancer cell lines (Figure 4) [123]. Whether TEAD autopalmitoylation inhibitors show activity against YAP-activated HNSCC remains an open question and an area of active investigation.

Numerous cellular mechanisms regulate YAP/TAZ activity. Thus, YAP/TAZ inhibition could be achieved through a variety of therapeutic avenues beyond the disruption of YAP/TAZ:TEAD interactions. As described earlier in this review, the GPCR-PKA signaling axis modulates Hippo-YAP/TAZ activity. GPCRs represent one of the most frequently targeted protein classes, with small molecule and peptide drugs designed to target essentially every type of GPCR either approved for clinical use or in clinical development [52,130]. Pharmacological inhibition of G_12/13_ or G_q/11_, or pharmacological activation of G_s_ could achieve downstream YAP/TAZ inhibition. Attempting to target the Hippo pathway far upstream of YAP/TAZ at plasma membrane bound receptors, however, could result in unforeseen effects [131,132].

PKA may represent another therapeutic target in GPCR-PKA-mediated regulation of the Hippo pathway. Interestingly, PKA inhibition via phosphatase PP2A activation has shown antineoplastic activity against models of small cell carcinoma [133]. Given that PP2A drives YAP/TAZ activation, the therapeutic use of PP2A phosphatase activators is unlikely to be beneficial in the treatment of HNSCC. However, drugs that target and inactivate PP2A may prove beneficial in the treatment of HNSCC. PP2A inhibitors are in current development and have shown efficacy against a variety of YAP-activated tumor types [134,135].

The interaction of β-catenin and YAP/TAZ also presents a potential therapeutically actionable opportunity. Wnt/β-catenin signaling has independently been associated with tumor initiation and progression and, as a result, multiple inhibitors of this pathway have been developed [136]. Given its role in downregulating β-catenin activity, the β-catenin destruction complex has been a focus for therapeutic intervention with the development of tankyrase inhibitors [132,133]. Tankyrase (TNKS) interacts with and degrades the β-catenin destruction complex component Axin via ubiquitin-mediated proteasomal degradation [137,138]. Tankyrase inhibitors and other compounds that stabilize Axin have been shown to inhibit both YAP/TAZ and β-catenin activity through the cytoplasmic retention of YAP/TAZ and subsequent restoration of destruction complex activity (Figure 4) [18,139]. In this manner, tankyrase inhibitors promote YAP/TAZ inhibition by the β-catenin destruction complex. More recently, tankyrase inhibitors have been shown to also exert a more direct effect on the Hippo pathway by stabilizing AMOT family proteins, which promote the cytoplasmic retention of YAP/TAZ and prevent YAP/TAZ induced transcriptional programs [124,140].

## 12. Conclusions

Since its discovery, intensive investigation of the Hippo pathway and its downstream effectors has considerably advanced our understanding of this dynamic signal transduction circuit and its ability to control normal tissue growth and cancer initiation and progression, therapy resistance, and metastatic spread. Inhibitors of YAP/TAZ activity, including newly developed TEAD autopalmitoylation inhibitors, hold significant promise as antineoplastic agents in multiple cancer types, including HNSCC. Given YAP and TAZ’s pleiotropic roles in tumor cell proliferation, cancer stem cell self-renewal and metastatic potential, YAP/TAZ inhibition is likely to display single agent activity in multiple cancer types that are dependent on YAP/TAZ function for tumor cell growth and survival. Yet the true potential of YAP/TAZ signaling inhibitors may be achieved by the combination with immune checkpoint inhibition, agents targeting additional oncogenic pathways, or the use of multimodal precision therapies co-targeting YAP/TAZ and their intrinsic and compensatory resistance pathways, thereby achieving durable cancer remission.

## Figures and Tables

**Figure 1 cells-11-01370-f001:**
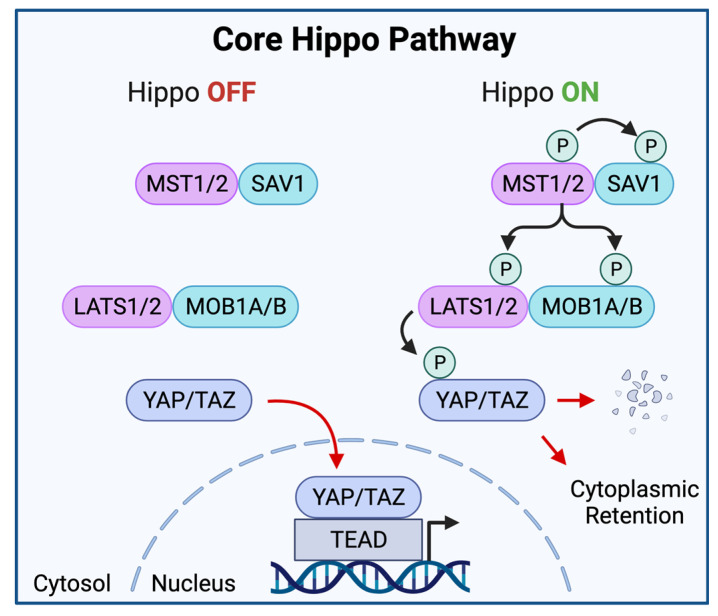
The Core Hippo Pathway. In the absence of Hippo-stimulating signals (**left**), the core Hippo kinase cascade is inactive and YAP/TAZ translocate into the nucleus to associate with TEAD family DNA binding proteins and co-activate transcriptional programs. In the presence of Hippo-stimulating signals (**right**), MST1/2 is phosphorylated and subsequently phosphorylates SAV1, LATS1/2, and MOB1A/B. In turn, LATS1/2 phosphorylates YAP/TAZ, resulting in YAP/TAZ cytoplasmic retention or proteolytic decay. Of note, Hippo-stimulating signals may bypass MST1/2 and directly activate LATS1/2.

**Figure 2 cells-11-01370-f002:**
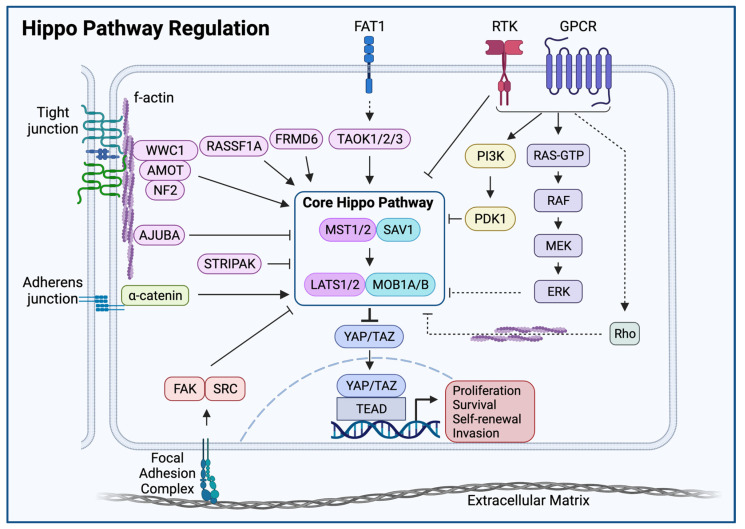
Hippo Pathway Regulation. A variety of mechanisms stimulate or inhibit core Hippo kinases including adherens and tight junctions, focal adhesions complexes, growth factor receptor tyrosine kinase (RTK) and G protein coupled receptors (GPCRs), and the actin cytoskeleton, as well as proto-cadherins of the FAT family. Solid arrows indicate evidence of direct interaction. Broken arrows indicate evidence of indirect or multistep interactions not detailed in this figure. Please see the text for further detail.

**Figure 3 cells-11-01370-f003:**
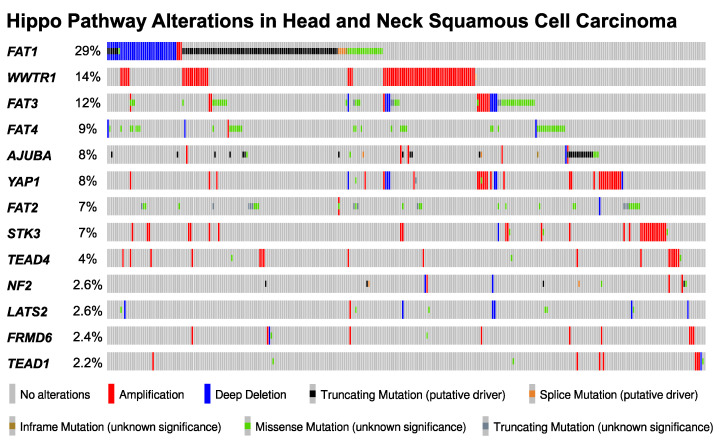
Hippo Pathway Alterations in Head and Neck Squamous Cell Carcinoma. Oncoprint illustration of the 13 most frequently altered Hippo pathway associated genes in HNSCC [76,77]. Genes are listed in accordance with the HUGO gene nomenclature committee (HGNC, www.genenames.org (accessed on 31 March 2022)): *WWTR1* denotes *TAZ*, *STK3* denotes MST2.

**Figure 4 cells-11-01370-f004:**
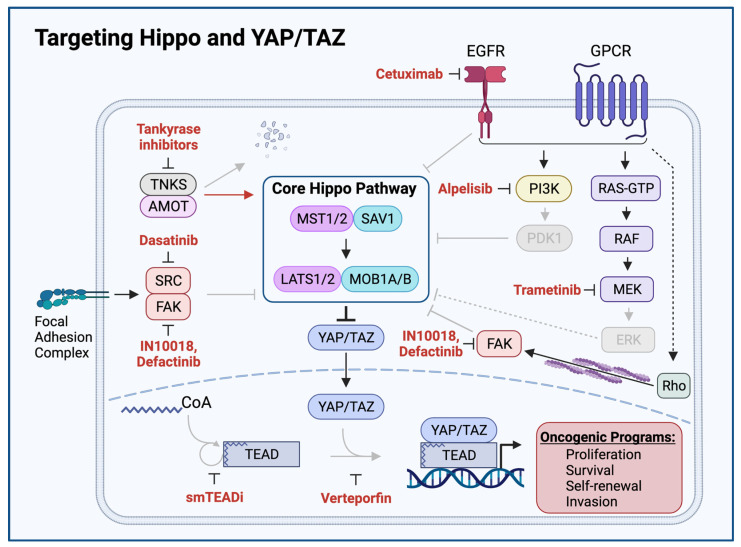
Targeting Hippo and YAP/TAZ. Therapeutic agents promoting Hippo activation or YAP/TAZ inhibition are shown in red. Grey arrows and molecules indicate inhibition of downstream processes. Cetuximab-mediated EGFR inhibition, Alpelisib-mediated PI3K inhibition, Dasatinib-mediated SRC inhibition, Defactinib and IN10018-mediated FAK inhibition, and Trametinib-mediated MEK inhibition relieve Hippo-inhibitory inputs and enable Hippo signaling to inhibit YAP/TAZ. Tankyrase inhibitors prevent proteolytic degradation of AMOT, enabling AMOT to activate Hippo signaling. Small molecule TEAD inhibitors (smTEADi) prevent YAP/TAZ:TEAD interaction by inhibiting TEAD autopalmitoylation. Verteporfin inhibits YAP/TAZ:TEAD interaction [11,116,120,123,124,125,126,127,128].

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
