# Peer review of "Genomic Hippo Pathway Alterations and Persistent YAP/TAZ Activation: New Hallmarks in Head and Neck Cancer"

_cells, 2022, doi:10.3390/cells11081370_

Round 1

Reviewer 1 Report

Dear authors, Farhoud Faraji * , Sydney I. Ramirez , Paola Y Anguiano Quiroz , Amaya N. Mendez-Molina , J. Silvio Gutkind *, thank you for your hard work. I am appreciative of such privilege to review your article. My recommendation is that the paper deserves minor revision before acceptance. I have documented this point below, which hopefully will help the authors to address the identified deficits.

  1. In page 4 and page 10, there shall be abbreviation for figures such as STRIPAK: striatin-interating phosphatase and kinase.

Author Response

We thank the reviewer for the kind words regarding this manuscript and for he helpful observation to include an abbreviations key for Figures 2 and 4. To address this point, we have appended an abbreviations key at the end of the Figure 2 legend. Given that all abbreviations in Figures 2 and 4 are shared, we believe that this addition will be adequate for both Figures 2 and 4.

Reviewer 2 Report

In this manuscript, authors discussed key elements of the mammalian Hippo pathway, detail mechanisms by which perturbations in Hippo signaling promote HNSCC initiation and progression, and outline emerging strategies to target Hippo signaling vulnerabilities as part of novel multimodal precision therapies for HNSCC.

The manuscript is well written and informative.

One minor point, whether there is any data link between Hippo pathway dysregulation and two carcinogenic etiologies underlie most HNSCC, such as papillomavirus (HPV) and chemical carcinogen exposure, need to be included in the discussion.  

Author Response

We thank the reviewer for this insightful suggestion. To address this reviewer’s point, we have added in the section entitled “FAT1: A Membrane-Associated Proto-Cadherin Assembling the Hippo Signalome” the following language:

  • “Interestingly, FAT1 mutation has been identified in carcinogen-induced models of HNSCC, potentially linking carcinogen-induced mutation and Hippo pathway pertur-bation as mediators of HPV-negative HNSCC initiation.” … “While inactivating FAT1 mutations are uncommon in HPV-positive HNSCC93, prelimi-nary analyses suggest that Hippo pathway perturbation and YAP/TAZ activation occur in HPV-positive HNSCC (not shown) and are an area of active investigation.”

In addition to addressing these specific changes, Reviewer 2 had also recommended carefully reading through the text to identify and correct minor grammatical and spelling errors, which we have performed and included in tracked changes in the revised manuscript. In addition, we have also updated Figures 2 and 4 where we found a typographical error in which “PDK1” was erroneously labeled as “PKD1”. Finally, we have updated our disclosures section.
